# Polyvinylidene Fluoride Membrane with a Polyvinylpyrrolidone Additive for Tofu Industrial Wastewater Treatment in Combination with the Coagulation–Flocculation Process

**DOI:** 10.3390/membranes11120948

**Published:** 2021-11-30

**Authors:** Irfan Purnawan, Derryadi Angputra, Septiana Crista Debora, Eva Fathul Karamah, Arifina Febriasari, Sutrasno Kartohardjono

**Affiliations:** Process Intensification Laboratory, Chemical Engineering Department, Faculty of Engineering, Universitas Indonesia, Kampus UI, Depok 1642, Indonesia; irfan.purnawan@ui.ac.id (I.P.); derryadi.angputra@ui.ac.id (D.A.); septiana.crista@ui.ac.id (S.C.D.); eva@che.ui.ac.id (E.F.K.); arifina.febriasari@ui.ac.id (A.F.)

**Keywords:** PVDF, PVP, tofu wastewater, ultrafiltration

## Abstract

Wastewater from the tofu industry contains many pollutants that are very harmful to the environment, significantly endangering aquatic life and producing a pungent odor. This study aims to prepare a polyvinylidene fluoride (PVDF) membrane with the additive polyvinylpyrrolidone (PVP), and utilize it to treat tofu wastewater in the ultrafiltration (UF) process. Flat sheet membranes were prepared using PVDF that was dissolved in *N*,*N*-dimethylacetamide (DMAc) and then combined with the additive material of PVP at the varying compositions of 14.9/0.1, 14.85/0.15, and 14.8/0.2 g of PVDF/gram of PVP. The addition of PVP was proposed to improve the properties of the membranes. Characterization by scanning electron microscope (SEM), water contact angle, and Fourier transform infrared spectroscopy (FTIR) were performed on the PVDF/PVP membrane flat sheet in order to understand and compare changes in the physical and chemical properties that occurred in the membrane. Prior to the UF process, the tofu wastewater was treated by a coagulation–flocculation process through a jar tester using poly aluminum chloride (PAC) as a coagulant. Based on the membrane characterization, the addition of PVP improved the physical and chemical properties of membranes. The pore size of the membrane becomes larger, which could increase permeability as well as the flux value. The TSS and turbidity of the water produced in the UF process decreased with an increase in feed pressure due to a greater driving force generated to facilitate the penetration of the suspended solids. The UF results showed that the effect of PVP on water flux was greatest for the 14.85/0.15 PVDF/PVP membrane for both pure and wastewater. In addition, the highest percentage of rejection for TSS and turbidity were observed in the 14.9/0.1 PVDF/PVP membrane and rejection for TDS was indicated in the 14.8/0.2 PVDF/PVP membrane. Meanwhile, the resulting pH decreased slightly across all samples as feed pressure increased.

## 1. Introduction

Tofu is one of the most prevalent foods in Asia, particularly in China, Japan, and Indonesia, and is recognized as a component of daily meals; it is a widely consumed and nutritious food. Tofu is simple, inexpensive, and contains many nutrients. It is made from soy, which is high in protein. Its manufacturing involves soybean grinding, boiling, protein coagulation, filtration, and preservation. Approximately 7–10 tons of liquid waste are produced for one ton of soybean used [1,2,3]. Every year, around two and a half million tons of soybeans are used by the tofu industries in Indonesia to produce tofu. As a result, each industry releases more than one million tons of solid waste and 20 million m^3^ of liquid waste annually [4,5,6,7]. The liquid waste contains soluble salts, carbohydrates, and substantial flavones, such as nitrogen, phosphorus, organic matter, and other nutrients, following the dissolution of these materials in the water during the production process [8,9,10,11,12]. Since most tofu industries are categorized as home industries to small-scale ones, the waste is often not treated correctly, and is disposed of freely into the surrounding water, thus damaging the environment and aquatic life [13]. Therefore, proper disposal is required to prevent the unpleasant smells that result from the accumulated waste that is known to endanger marine biota and pollute underground water sources. Furthermore, tofu wastewater is rich in organic content with a Biological Oxygen Demand (BOD) and Chemical Oxygen Demand (COD) range of 6000–8000 mg/L and 7500–26,000 mg/L, respectively, based on the corresponding quality standards of 150 and 275 mg/L, respectively [14,15]. The tofu wastewater needs to be treated to reduce the dangerous impact on the environment, particularly the aquatic environment. Various methods are used to treat both types of waste, ranging from simple techniques, namely coagulation and precipitation [16], to more complex methods that use filters such as membranes [17,18].

Numerous types of membrane materials have been utilized in the various separation processes. Polymer-based membranes are currently the most frequently utilized membranes for many purposes, including microfiltration (MF), ultrafiltration (UF), nano-filtration (NF), and reverse osmosis (RO) [19]. Of all polymeric materials, such as polysulfone (PSF), polypropylene (PP), polytetrafluoroethylene (PTFE), polyethylene (PE), and polyvinylidene fluoride (PVDF) [20,21], the PVDF membrane is the most extensively employed membrane used to fabricate UF and MF membranes since it has outstanding mechanical, physical, and chemical properties, as well as excellent thermal stability [22,23]. Although PTFE has a similar structure and advantages when compared to PVDF, PTFE has a high density whereas PVDF has a low density and higher porosity, which generally results in a higher water flux [24]. In addition, the PVDF membrane has the capacity to separate organic matter, including carbohydrates, proteins, and fats, and also possesses a high oxidant tolerance and great mechanical strength with excellent resistance to fouling [19]. Furthermore, many organic solvents can be used to dissolve PVDF, such as *N*,*N*-dimethylacetamide (DMAc), which is used in this study. The phase-immersion precipitation technique is often applied for membrane preparation, where the base material, solvent, and non-solvent are mainly required. However, a combination of PVDF with *N*,*N*-dimethylacetamide (DMAc) yields a more powerful solvent compared to others, including tetraethyl phosphate (TEP), *N*-methyl-2-pyrrolidone (NMP), and dimethylformamide (DMF) [25]. The polar nature of DMAc enables it to act as a combined solvent and reaction catalyst in many reactions, producing high yields and a pure product in short time periods. It is also a versatile solvent due to its high boiling point and good thermal and chemical stabilities [26]. Previous studies have investigated the effects of three type of solvents (DMAc, DMF, DMSO) on the membrane porosity, mechanical properties, and pure water flux of a PVDF membrane; the addition of a substance such as PVP was found to increase the surface hydrophilicity of the PVDF membrane [27]. Based on previous studies, PVP can enhance the porosity of the PVDF membrane and improve the pure water flux [28]. PVP is also recognized as a nontoxic substance and has decent solubility in water and other solvents, making PVP a suitable polymer additive [29,30,31,32]. This study aims to prepare and characterize PVDF membranes with PVP as an additive to increase the hydrophilicity of the PVDF membrane for tofu wastewater treatment. Based on previous research on hollow fibre membranes, the combination of PVDF with PVP exhibits good structure, excellent mechanical properties, and high porosity [33]. Therefore, it is expected that the addition of PVP to the PVDF membrane could improve the hydrophilicity and surface properties of the mixed membrane.

## 2. Materials and Methods

### 2.1. Membrane Preparation

The polyvinylidene fluoride (PVDF) was purchased from Solvay Chemical USA, *N*,*N*-dimethylacetamide (DMAc) was purchased from Merck Indonesia, and polyvinylpyrrolidone (PVP) was purchased from Sigma-Aldrich Indonesia. Meanwhile, ethanol and deionized water which were purchased from PT. Dwinika Intan Mandiri Indonesia. The composition of the casting solution is the independent variable in this research. Dependent variables are casting time, stirring speed, and coagulation temperature. The composition of each casting solution is presented in Table 1.

The phase inversion method is generally used for flat sheet membrane preparation [19]. The PVDF membrane was prepared by dissolving it in DMAc with the addition of PVP as an additive. The composition of PVDF, DMAc, and PVP is shown in Table 1. DMAc was first put in an Erlenmeyer flask, and then the PVDF and PVP were slowly added at 25 °C and 200 rpm stirring. The dissolution process was successive by increasing the temperature to 60 °C and the stirring speed to 300 rpm until the polymer and additives were completely dissolved. The stirring continued with a speed of 500 rpm for about 3 h until the solution was homogenous. The homogeneous PVDF/PVP solution, which is called the casting solution, was then allowed to stand at ambient temperature for the bubble-removal process until the foam that had formed has been removed. The casting solution was cast on to glass film using a casting-glass roller, and then immersed in deionized water for about 24 h. Afterward, the formed flat sheet membrane was soaked in a 96 wt. % ethanol solution for 30 min and 50 wt. % ethanol solution for 1 h before being dried in the open air at room temperature to be characterized.

Several methods were used to characterize any changes in the physical and chemical properties of the membrane, such as scanning electron microscopy (SEM), measuring the water contact angle, and Fourier transform infrared spectroscopy (FTIR). The surface hydrophilicity of PVDF and PVP/PVDF membranes was measured by water contact angle (CA) at ambient temperature. The membrane was placed on a solid surface, and the deionized water, as a liquid probe for each measurement, was dropped onto the membrane surfaces. The results were obtained using a high-resolution camera to evaluate the angle between the solid and the liquid. Fourier transform infrared spectroscopy (FTIR) analysis was performed to confirm the occurrence of the PVP additive in the PVDF membrane. To obtain the value for the membrane porosity, the following formula was used [34]: (1)$$\varepsilon \;\left(\%\right)=\frac{W_{w}-W_{d}}{\rho_{w}.A.l}\times 100$$
where *W_d_* is the weight of the dry membrane, *W_w_* is the weight of the wet membranes that was boiled in water for 2 h and then allowed to sit in water for 24 h at room temperature. *ρ*_w_ is the density of water, *A* is the effective membrane area, and *l* is the membrane thickness. 

The following equation was used to calculate the average membrane pore size using the Guerout–Elford–Ferry formula through the filtration velocity method [35].
(2)$$r_{m}=\sqrt{\frac{\left(2.9-1.75\varepsilon \right)\times 8\eta lQ}{\varepsilon \times A\times ∆P}}$$
where *ε* is the membrane porosity, *Q* is the volume of permeate per unit time (m^3^.s^−1^), *l* is the membrane thickness, *η* is the viscosity of pure water (8.9 × 10^−4^ Pa. s), *A* is the effective surface area of the membrane, and Δ*P* is the transmembrane pressure.

### 2.2. Instrumentation for Membrane Characterization

Scanning electron microscopy (SEM, ZEISS Ultra 60) was utilized to examine the surface and cross-section morphologies of the PVDF and PVP/PVDF membranes. The specimens were coated with a thin layer of gold, and the thin morphology layer was viewed at 20.0 kV of excitation voltage. The membrane samples were frozen in liquid nitrogen and fractured before the surface and cross-sectional images were taken in order to avoid unnecessary stress on the membranes. Finally, the membrane was sprayed with a thin layer of gold for electron conductivity [36]. Thermo Scientific FTIR (Diamond Nicolet IS 5) was utilized to examine the surface chemical compositions of the PVDF and PVP/PVDF membranes. The spectra were recorded at a resolution of 32 cm^−1^ by 64 scans in the range of 500–4000 cm^−1^.

### 2.3. Pretreatment of Tofu Wastewater

The tofu wastewater pretreatment process was needed because the suspended solids and turbidity can interfere with the ultrafiltration process. The tofu wastewater was treated before the ultrafiltration process by a coagulation–flocculation process through a jar tester consisting of six batch stir reactors with identical stirring conditions. The pH of the wastewater was adjusted to around 7 by adding a 2M NaOH solution before the coagulation–flocculation process. First, 3 mL of 5% PAC (Poly Aluminum Chloride) solution was added to 500 mL of tofu wastewater so that the concentration of PAC in the wastewater was around 300 ppm [37]. The coagulation–flocculation process is conducted by fast stirring at a speed of 120 rpm for 2 min, followed by slow stirring at a speed of 40 rpm for 10 min. The tofu wastewater from the coagulation–flocculation was filtered through a vacuum filter using filter paper with a pore size of 10 µm and then settling for 30 min.

### 2.4. Ultrafiltration Process

The membrane was cut into circles measuring 4.5 cm in diameter and placed in a permeation test cell to determine the flux and solute rejection. Table 2 shows data for the pre-treated wastewater fed into a 1 L pressurized tank prior to the experiments, as represented in Figure 1. The feed pressure for the UF process was provided by pressurized nitrogen gas supplied to the feed tank. The independent variables include the composition of the PVP in the PVDF casting solution and the feed pressure used in the UF process. In contrast, the dependent variables are water flux, pH, and TSS, TDS, and turbidity rejections. Specifically, turbidity and TSS were measured using a Colorimeter DR/890, while pH and TDS were determined by a Hanna Combo pH meter, and water flux was obtained by evaluating the permeate volume collected for 2 min from the following equation [38,39]:(3)$$J=\frac{V}{A\times ∆t}$$
where *J*, *A*, and *V* are the permeate flux (L.m^−2^.h^−1^), the cross-sectional area of the membrane (m^2^), and the collected volume (L), respectively, at a time interval Δ*t* (h^−1^). Rejection values were calculated based on the concentrations of feed and permeate using the equation [40]:(4)$$\%R=\left(\frac{C_{f}-C_{p}}{C_{f}}\right)\times 100\%$$
where *C_f_* and *C_p_* are the concentration of solutes in feed and permeate streams, respectively.

## 3. Results and Discussion

### 3.1. Membrane Characterization

Figure 2 shows the results of the characterization using FTIR for whole membrane samples from only PVP, only PVDF, and PVDF/PVP with various compositions of PVP addition. Peaks at wave numbers 840 cm^−1^ and 1250 cm^−1^ show CF_2_ stretching and CH_2_ bonding, which belong to PVDF. As demonstrated in Figure 2, there is a deep valley in PVP at a wavelength of 1627 cm^−1^, while there is no valley in PVDF at around the same wavelength. Very shallow valleys can be seen in other curves for PVDF with the various additions of PVP at wavelengths between 1631–1641 cm^−1^, proving that PVP exists in the blended membranes. This phenomenon shows the stretching of the carbonyl bond in PVP. The peak appears very small because the concentration of PVP added is quite low, i.e., PVDF/PVP 14.9/0.1 to 14.8/0.2 (0.67 to 1.35% of PVDF weight). In addition, it shows that there is no chemical interaction between PVDF and PVP. 

The contact angle characterization results, as presented in Table 2, show that PVDF Pristine has a water contact angle of 73.45°, while PVDF/PVP 14.8/0.2, PVDF/PVP 14.85/0.15, and PVDF/PVP 14.9/0.1 have water contact angle values of 55°, 61°, and 66°, respectively. All PVDF membranes with various PVP additions have water contact angle values below 90°, as shown in Table 2, which suggests they all have hydrophilic properties [41]. From these results, the higher the composition of PVP, the lower the value of the contact angle, which suggest that the addition of more PVP results in the membrane surface being more hydrophilic. Similar results were previously reported, where the addition of nanoparticle PVP to the PVDF membrane can increase the hydrophilicity of the PVDF membranes due to the super hydrophilic PVP molecules, which affect the interactions between water molecules and the membrane surface [31,42]. Differences in membrane hydrophilicity can also be caused by differences in surface morphology, which can depend on the phase inversion process [29].

The SEM characterization for the membrane surface is shown in Figure 3, and the SEM characterization for the membrane cross-section is shown in Figure 4. Compared with pure PVDF (without PVP), the membrane with PVP can open or widen the pores of the PVDF membrane, as presented in Figure 3. As can be seen from Figure 3, the most extensive pore size range is on the PVDF/PVP 14.85/0.15 membrane, while the smallest pore range is on the PVDF/PVP 14.9/0.1 membrane. However, this observation still needs to be confirmed with the value of water flux on each membrane through experiments with water or wastewater samples. 

The effect of the PVP additions on thickness, porosity, and pore size can be seen in the measurement results in Table 3. As can be seen from Table 3, PVDF pristine (a) has a thickness of 65.23 µm, while PVDF/PVP 14.8/0.2, PVDF/PVP 14.85/0.15, and PVDF/PVP 14.9/0.1 have a thickness of 77.78 µm, 51.11 µm, and 139.8 µm, respectively. From the SEM photo results, the most impenetrable membrane is PVDF/PVP 14.9/0.1, while the thinnest membrane is PVDF/PVP 14.85/0.15. In the PVDF/PVP 14.9/0.1 membrane, the addition of PVP has a kinetic effect compared to a thermodynamic effect, which increases the viscosity of the dope solution. It caused the PVDF/PVP 14.9/0.1 membrane to have a greater thickness. Meanwhile, based on the results of porosity measurements, the addition of PVP to the PVDF/PVP 14.85/0.15 and PVDF/PVP 14.8/0.2 membranes gave a more dominant thermodynamic effect than a kinetic effect. With the thermodynamic effect, PVP can accelerate the change of solvent-non-solvent during the membrane coagulation process, increasing the membrane porosity, although the average pore size is smaller [28]. The average pore size of the pristine PVDF membrane could not be identified as the membrane was dense, so the permeate value per unit time cannot be determined.

### 3.2. Pretreatment Process

Table 4 shows the characteristics of the membrane before and after the pretreatment process. The results of the pretreatment of tofu wastewater had a significant impact on the reduction of TSS and turbidity. It reveals that the coagulation–flocculation process makes the suspended solids unstable and form macro flocs that can be separated from the wastewater solution using filter paper [13]. Meanwhile, the TDS of wastewater after pretreatment increased due to the addition of NaOH during the pH adjustment.

### 3.3. Separation Process

The deionized water and pre-treated wastewater fluxes determined the PVDF membrane performance in the UF process. Figure 5 shows the pure water flux as a function of the feed pressure in the UF process. Meanwhile, Figure 6 presents the wastewater flux as a function of the feed pressure in the UF process. The formation of the porous polymer was aided by the PVP, known to be a hydrophilic additive. Therefore, adding more PVP to the casting solution tends to enlarge the membrane porosity. As presented in Figure 5, a simultaneous increase in water flux following a rise in the feed pressure and greater PVP composition was due to a higher driving force and expanded membrane, respectively [43]. Also, the pure water flux expanded from 64.2 to 176.2 L.m^−2^.h^−1^, 93.8 to 213.7 L.m^−2^.h^−1^_,_ and 171.6 to 396.8 L.m^−2^.h^−1^ for PVDF/PVP 14.9/0.1, 14.85/0.15, and 14.8/0.2 membranes, respectively, after the upsurge in feed pressure from 4 to 7 bar. The same phenomenon was also reported by Beygmohammdi and colleagues [31], where the addition of PVP to the PVDF membrane increased the flux of pure water.

As shown in Figure 6, the change in feed pressure instigates an increase in water flux due to a higher driving force. The highest value occurred in the 14.85/0.15 PVDF/PVP membrane as opposed to the 14.8/0.2 PVDF/PVP membrane observed for the pure water. It indicates that membrane thickness is a more dominant factor in wastewater flux than membrane porosity. The water flux increased from 26.2 to 76.9 L.m^−2^.h^−1^, 45.7 to 160.4 L.m^−2^.h^−1^, and 20.1 to 115.1 L.m^−2^.h^−1^ for PVDF/PVP 14.9/0.1, 14.85/0.15, and 14.8/0.2 membranes, respectively, after an increase in feed pressure from 4 to 7 bar.

Figure 7 and Figure 8 represent TSS and turbidity rejections for wastewater, respectively, with a maximum value observed for the 14.9/0.1 PVDF/PVP membrane. The 14.9/0.1 PVDF/PVP membrane shows the lowest porosity compared to others, and thus does not allow penetration. Generally, the rejections decreased following an increase in feed pressure, resulting from the cumulative driving force assumed to facilitate the penetration of suspended particles and turbidity through the membrane [28]. Based on these results, the TSS and turbidity rejection decreased from 93.3 to 26.7%, 86.8 to 8.8%, and remained relatively constant at 94.1% for the PVDF/PVP 14.8/0.2, 14.85/0.15, and 14.9/0.1 membranes, respectively, with an increasing feed pressure from 4 to 7 bar. The decrease in TSS and turbidity was almost the same as the main constituent of turbidity was the suspended solids in tofu wastewater [13].

Figure 9 shows TDS rejection of tofu wastewater as a function of the feed pressure in the UF process. Meanwhile, Figure 10 presents the pH of water produced in the UF process at various feed pressures. TDS measures solids, which includes ions, compounds, and colloids dissolved in water. Figure 9 shows that the UF process is not suitable for evaluating reduction rates ranging from 3.7% to 19.9% [44]. TDS rejection increased when the feed pressure was increased to 7 bar, indicating that the membrane prevents access to dissolved particles. Subsequent increases ranging from 13.4 to 19.9%, 11.6 to 19.6%, and 3.7 to 10.4% for PVDF/PVP 14.8/0.2, 14.85/0.15, and 14.9/0.1 membranes, respectively, reflects an increase in feed pressure from 4 to 7 bar.

Figure 10 revealed that the pH of permeated water slightly decreased after an increase in feed pressure, showing no significant effect. In addition, Figure 10 further indicates that a more porous membrane produces water with a higher pH due to the other suspended solids left behind [44]. However, the pH of permeated water decreased from 8.08 to 7.98, 7.84 to 7.42, and 7.02 to 6.79 for the PVDF/PVP 14.8/0.2, 14.85/0.15, and 14.9/0.1 membranes, respectively, after an increase in feed pressure from 4 to 7 bar.

## 4. Conclusions

Characterization by scanning electron microscope (SEM), water contact angle, and Fourier transform infrared spectroscopy (FTIR) were conducted on PVDF/PVP flat sheet membranes to understand and compare changes in the physical and chemical properties that occurred in the membrane. The addition of PVP to the PVDF membrane benefits the membrane in achieving better features than PVDF alone. According to the FTIR test, the presence of PVP is confirmed by a shallow valley at a wavelength in the range of 1631–1641 cm^−1^ in the PVDF membrane line curve, which has no valley at the same wavelength. The water contact angle characterization shows that with the higher composition of PVP, the lower the contact angle measured, which indicates a more hydrophilic membrane surface. The addition of PVP improves the contact degree of the water contact angle due to the enlargement of the membrane pores. SEM characterization was employed to both the surface and a cross-section of the membranes. Compared with pristine PVDF, the membranes with PVP have larger pores, with the most extensive pore size range belonging to the PVDF/PVP 14.85/0.15 membrane, which is between 51.6–207.6 nm. In addition, the membranes with the addition of PVP are thicker, with the thickest being the PVDF/PVP 14.9/0.1 membrane at about 139.8 µm. As supported by the data, the addition of PVP provides enhanced properties to the PVDF membrane. The TSS and turbidity of water produced in the UF process decreased with an increase in feed pressure due to a greater driving force generated to facilitate the penetration of suspended solids. The rejections of TSS, turbidity, and TDS were 94.1%, 93.0%, and 19.9%, respectively, at a feed pressure of 7 bar. The experimental results showed that the decrease in TSS was almost the same as the decrease in turbidity due to the main constituent of turbidity being suspended solids in the tofu wastewater. Meanwhile, the feed pressure applied did not significantly affect the pH of the water produced. Based on the results of the rejection, the majority of the optimal rejection values belong to the PVDF/PVP 14.9/0.1 membrane. Therefore, it can be concluded that the optimum porosity for the application in this study is 32%.

## Figures and Tables

**Figure 1 membranes-11-00948-f001:**
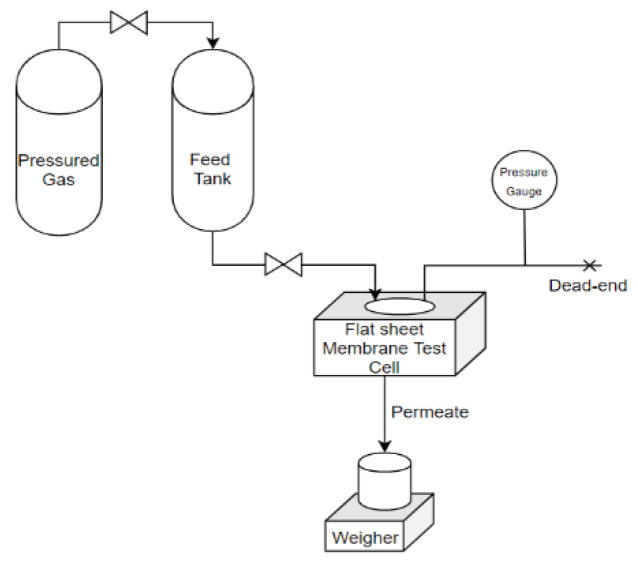
Illustration of the ultrafiltration process.

**Figure 2 membranes-11-00948-f002:**
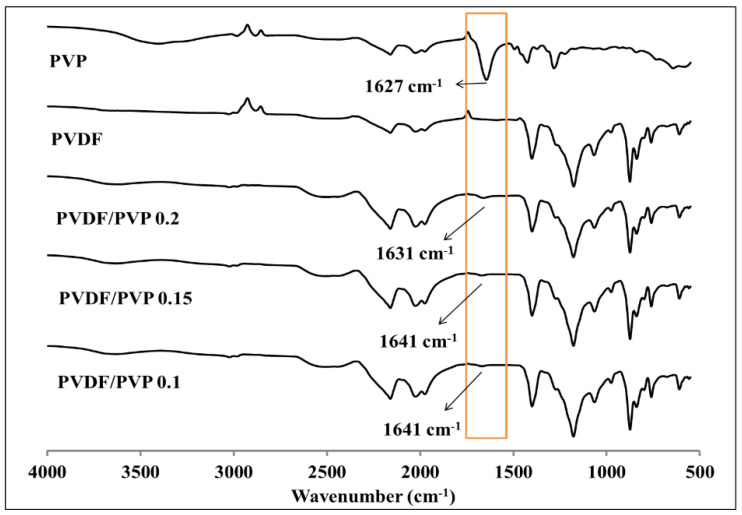
FTIR spectra of PVP, PVDF membranes, and PVDF membranes with various additions of PVP.

**Figure 3 membranes-11-00948-f003:**
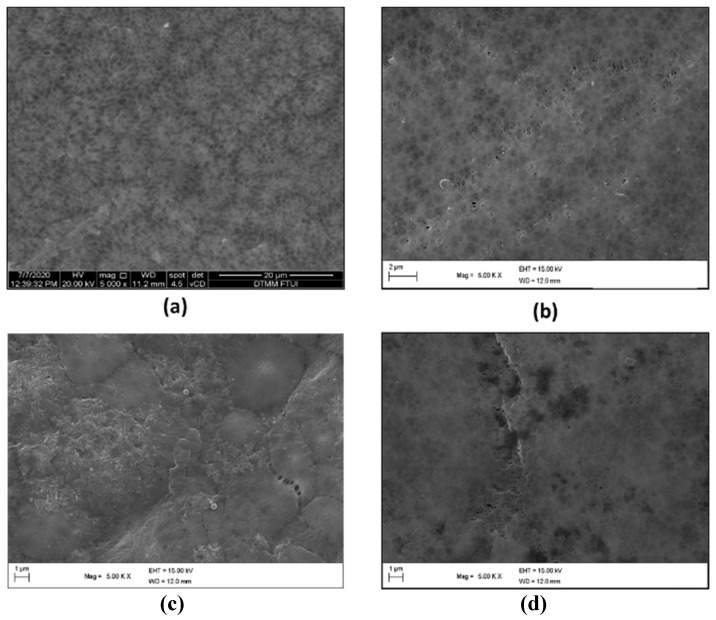
SEM results for the membrane surface of (**a**) PVDF Pristine, (**b**) PVDF/PVP 14.8/0.2, (**c**) PVDF/PVP 14.85/0.15, and (**d**) PVDF/PVP 14.9/0.1.

**Figure 4 membranes-11-00948-f004:**
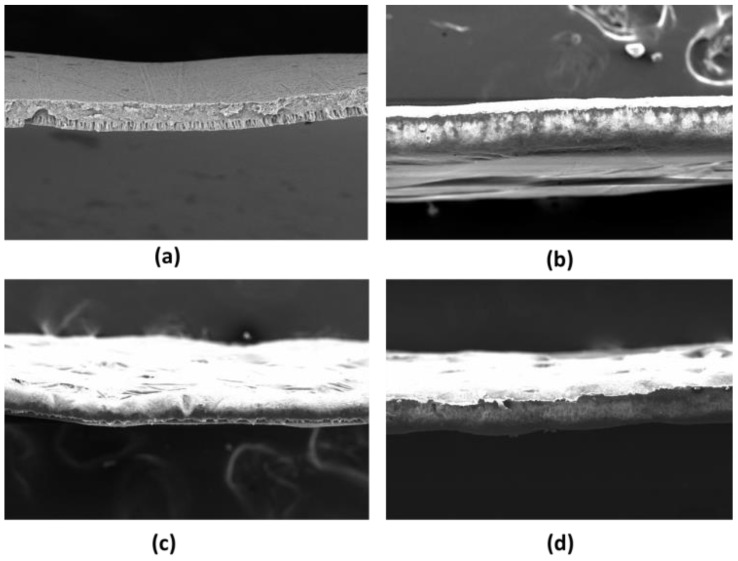
SEM results for the membrane cross section of (**a**) PVDF Pristine, (**b**) PVDF/PVP 14.8/0.2, (**c**) PVDF/PVP 14.85/0.15, and (**d**) PVDF/PVP 14.9/0.1 with a 100× magnification.

**Figure 5 membranes-11-00948-f005:**
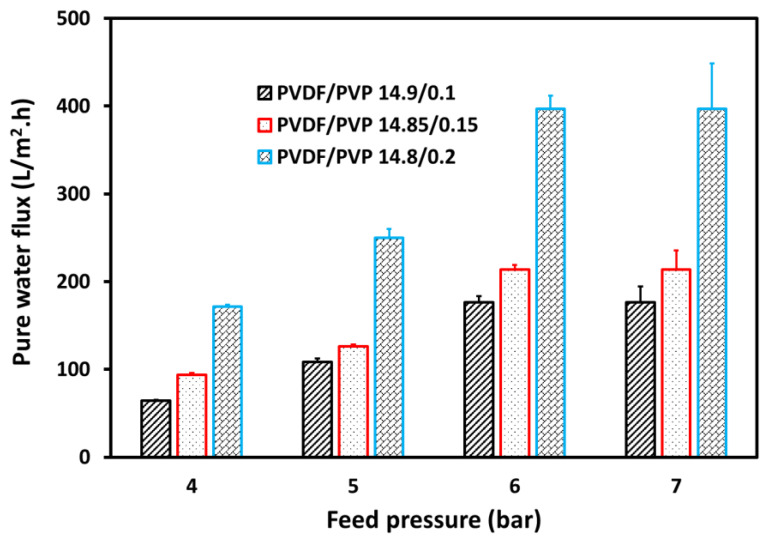
Water flux at various feed pressures in the UF process for pure water.

**Figure 6 membranes-11-00948-f006:**
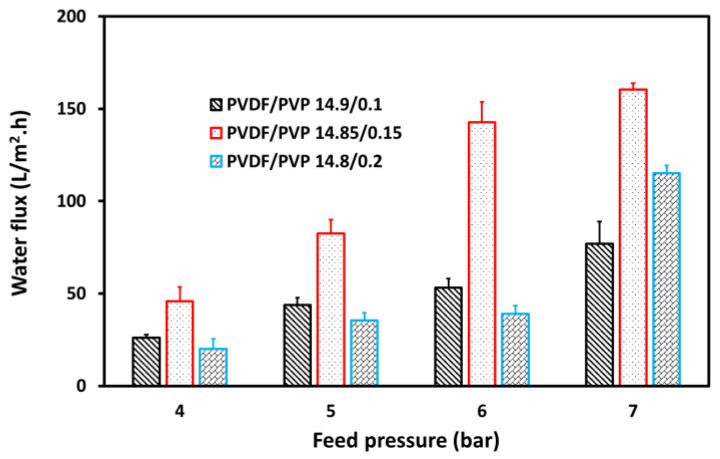
Water flux at various feed pressures in the UF process for tofu wastewater.

**Figure 7 membranes-11-00948-f007:**
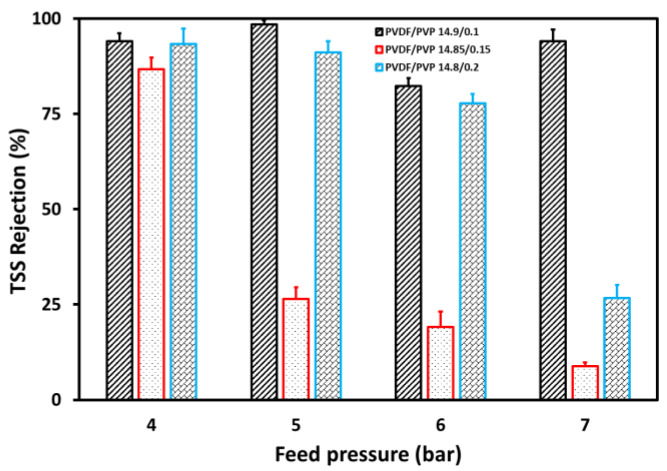
TSS rejection at various feed pressures in the UF process for tofu wastewater.

**Figure 8 membranes-11-00948-f008:**
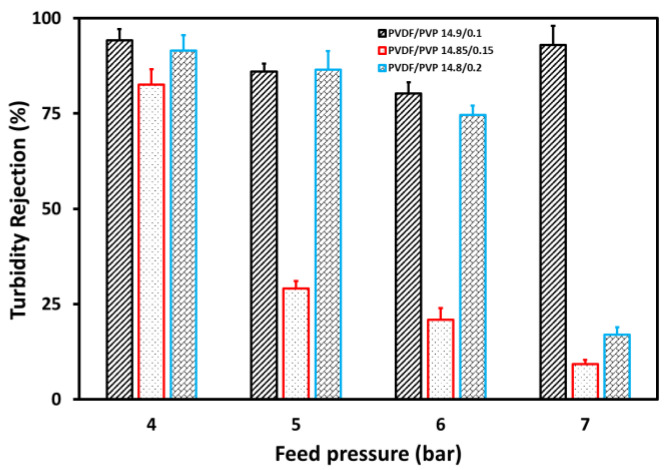
Turbidity rejection at various feed pressures in the UF process for tofu wastewater.

**Figure 9 membranes-11-00948-f009:**
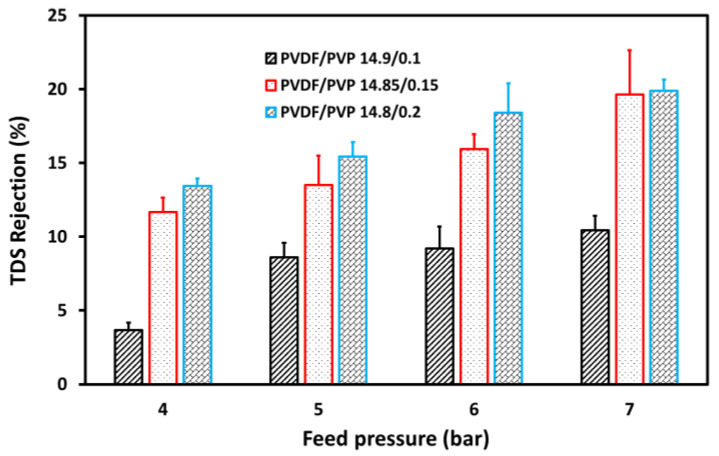
TDS rejection at various feed pressures in the UF process for tofu wastewater.

**Figure 10 membranes-11-00948-f010:**
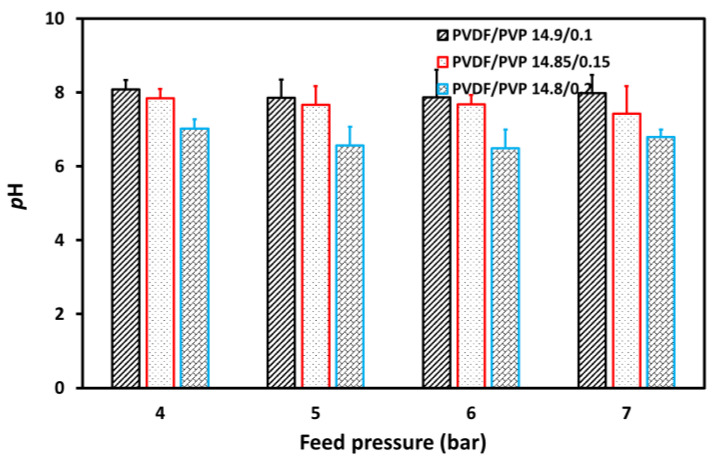
The pH of permeated water at various feed pressures in the UF process for tofu wastewater.

**Table 1 membranes-11-00948-t001:** Composition of the casting solutions.

PVDF (g)	DMAc (mL)	PVP (g)	Code
15	85	-	PVDF
14.9	85	0.1	PVDF/PVP 0.1
14.85	85	0.15	PVDF/PVP 0.15
14.8	85	0.2	PVDF/PVP 0.2

**Table 2 membranes-11-00948-t002:** Water contact angle of PVDF with various additions of PVP.

Parameter	Water Contact Angle
PVDF Pristine	73.45° ± 0.5°
PVDF/PVP 14.8/0.2	55° ± 2.1°
PVDF/PVP 14.85/0.15	61° ± 2.3°
PVDF/PVP 14.9/0.1	66° ± 3.5°

**Table 3 membranes-11-00948-t003:** Thickness, porosity, and average pore size of the membranes.

Membrane	Thickness (µm)	Porosity (ε, %)	Avarage Pore Size (nm)
PVDF Pristine	65.23	9.23 ± 1.5	Un-identified
PVDF/PVP 0.1	139.8	32.01 ± 0.4	14.9
PVDF/PVP 0.15	51.11	79.41 ± 0.9	6.7
PVDF/PVP 0.2	77.78	93.59 ± 1.3	5.9

**Table 4 membranes-11-00948-t004:** Tofu wastewater data before and after pre-treatment.

Parameter	Before Pre-Treatment	After Pre-Treatment
pH	4.01	6.78 ± 0.21
TSS (mg/L)	501	56.5 ± 1.2
TDS (mg/L)	833	1820 ± 190
Turbidity (FAU)	594	72.5 ± 2.75

## Data Availability

The data presented in this study are available upon request from the corresponding author. The data are not publicly available due to restrictions from the tofu industry.

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
