# Peer review of "Polyvinylidene Fluoride Membrane with a Polyvinylpyrrolidone Additive for Tofu Industrial Wastewater Treatment in Combination with the Coagulation–Flocculation Process"

_membranes, 2021, doi:10.3390/membranes11120948_

Round 1

Reviewer 1 Report

Dear Authors, below are my comments, please answer and incorporate in the text. 

The membrane you have prepared had different thicknesses? did you consider this in your results' analyses?

The porosity and flux increased, did you test the purity of the permeate?

You did not examine PVDF Pristine, this can be a control? This is very important to include?

I dont see novelty in your membrane preparation approach, please justify?

Figure 1 and carrying out the experiment are not clear, put more effort in describing the experiment.

Figure 4 is not good in quality!

What is the optimum porosity and void gap size for such application?

Have you done some physical analysis for the new membrane? what are the results if any?

We dont have the results against time? what was the permeate flux agaisnt time?

Reviewer 2 Report

The title is clear, please don’t use abbreviation in title, aren’t necessary.

The subject matter is within the scope of the journal.

This article contains new aspects, but the authors must underline the major findings of their work, and explain how yours study represent a progress comparatively with other papers. Please clearly explain the novelty.

The manuscript is easy to understand by scientists form other area.

The Abstract must be revised. The Abstract section not refers to the study findings, methodologies, discussion as well as conclusion.

The key words permit found article in the current registers or indexes.

Please complete Citation on the first page.

In the introduction it is clearly described the state of the art of the investigated problem.

The Materials and Methods can be improved. For example the characteristics of devices used in this study must be presented. I don’t find type of SEM… for example.  

Please verify all units of measurement… there are wrong type.

The figures have relatively good quality.

In Tables are presented necessary results.

Please verify:

3.2. Pretreatment process

Table 4 shows the characteristics of the membrane before and after the pretreatment process.

Water is pretreated not membranes. It is confuse…

Comparisons with other researches are necessary.

The literature is sufficiently critical, current, and internationally evaluated.

The size of the article is appropriate to the content.

The conclusions are OK

Please citation references in correct format.

For the references:

Huang, F. Y. C.; Arning, A., Performance Comparison between Polyvinylidene Fluoride and Polytetrafluoroethylene Hollow Fiber Membranes for Direct Contact Membrane Distillation. 2019, 9 (4), 52.

and

Fontananova, E.; Jansen, J. C.; Cristiano, A.; Curcio, E.; Drioli, E. J. D., Effect of additives in the casting solution on the  formation of PVDF membranes. 2006, 192 (1-3), 190-197.

missing journal (Membranes, respectively Desalination)

Reference:

Barambu, N. U.; Bilad, M. R.; Bustam, M. A.; Kurnia, K. A.; Othman, M. H. D.; Nordin, N. A. H. M., Development of membrane material for oily wastewater treatment: A review. Ain Shams Engineering Journal 2020.

Was published in 2021: volume 12, Issue 2, Pages 1361-1374

The paper was written in standard, grammatically correct English. Small corrections are necessary.

The guide for author is not respected.

Please complete:

Author Contributions:

Data Availability Statement: 

MAJOR REVISION in this form

Reviewer 3 Report

The manuscript is presenting an interesting piece of work having an important meaning for the industry.

Abstract: It will be much easier to follow if the ratio PVP to PVDF will be presented not the mass of the additive material.

The introduction section is organized in a good way, however, the novelty of the work should be better highlighted.

In the material part please add the country of origin of the materials and chemicals.

It will be more readable if the applied scientific equipment will be presented in a separate section not together with the method of casting membranes.

Results and discussion: Fig.2 – why there are bands in different directions? Was it related to the spectrum normalization? Please improve the quality of the spectra. A deeper interpretation of the FTIR spectra should be provided.

Please take care of accuracy, in FRIT and goniometric data presentation (Table 2).

Line 202 – from where the pore size was taken? What type of equipment was applied with such huge accuracy to pm??

The same magnification of the SEM images should be presented to have a chance for comparison. The membranes for cross-section Fig. 4b,c,d. were prepared in a wrong way. There were no cut in the liquid nitrogen. This must be repeated.

Table 4 – please check if really TDS was so much higher after the process?

The discussion in the separation process section should be more scientific and more critical. Please compare the established data with the data available in the literature and compare them.

The English should be improved in the manuscript.

Although the work is an interesting quality of the manuscript, data presentation should be improved.

Round 2

Reviewer 1 Report

Accept

Author Response

We thank you very much for the valuable review.

Reviewer 2 Report

The manuscript was improved in accord with recommendations.

Author Response

(The authors gave the same response as above.)

Reviewer 3 Report

The manuscript has been partially corrected but still, there are points that were not taken into account. 

Fig. 2 - take care on the quality, no need to present 2 digits after the comma. It looks unprofessional. moreover, the spectra look squised. 

Fig. 4 - the images are not corrected. It was pointed out that samples were prepared in the wrong way. This must be corrected and analysis should be repeated. To present a good quality cross-section of the membranes. 

If the pore size were determined only from the SEM this is not correct. In that case, only the surface porosity of the material can be presented and discussed. The analysis, e.g. porosimetry should be applied to determine the pore size with higher accuracy. Furthermore, there is no information about the active pores in the transport/separation process. And this information is crucial. The authors should add the pore size, pore size distribution analyses. 

line 362 - too much accuracy in the presented results. 

Author Response

Please see the attachmnet.

Round 3

Reviewer 3 Report

The manuscript has been corrected and is ready to be published.